# Economic Sustainability, Innovation, and the ESG Factors: An Empirical Investigation



Luca Di Simone [1,*] , Barbara Petracci [2] and Mariacristina Piva [3]

1    Department of Economic and Social Sciences, Università Cattolica del Sacro Cuore, 29122 Piacenza, Italy
2    Department of Management, University of Bologna, 40126 Bologna, Italy; barbara.petracci@unibo.it
3    Department of Economic Policy, Università Cattolica del Sacro Cuore, 29122 Piacenza, Italy;
     mariacristina.piva@unicatt.it
*    Correspondence: luca.disimone@unicatt.it

**Abstract:** The growing attention to sustainability has generated increasing interest in its relevant determinants and a possible relationship with economic growth's main drivers. Our paper contributes to this literature in three ways, by proposing the following empirical analysis of most innovative companies listed worldwide (909 firms over the 2013–2017 time-span): firstly, market-perceived innovation—proxied by the interaction between R&D intensity and the market-to-book ratio—has a positive impact on economic sustainability; secondly, when the three ESG pillars are considered, the social one turns out to have the highest effect on economic sustainability; thirdly, results are confirmed even when we control for context-specific conditions.

**Keywords:** economic sustainability; innovation; R&D expenditures; ESG factors; listed companies; panel analysis

**JEL Classification:** O30 (General-Innovation); M14 (Corporate Culture-Diversity-Social Responsibility); Q01 (Sustainable Development)

## 1. Introduction

Motivated by the increasing worldwide attention to the "sustainability" pathway (see the United Nations Sustainable Development Goals (SDGs)), the purpose of this work is an empirical investigation of the relationship between innovation and economic sustainability at the corporate level. In a specific way, our work attempts to answer the following fundamental question: does innovation affect economic sustainability or not? Are these two dimensions connected, or are they disjoint? The answer is not trivial, and evidence might have significant implications both at the individual and aggregate terms. If innovation positively affects economic sustainability, it is plausible to imagine a significant evolution based on specific optimal combinations; otherwise, the interpretation is a structural antithesis between them, implying trade-offs and unpopular decisions.

Since the beginning of the new millennium, sustainability has received strong consideration in the wake of several cross-cutting issues. Relevant but certainly not exhaustive examples of a renewed interest in sustainability topics are linked to climate change, growing industrial pollution, accounting and financial scandals (frauds), and managers' excessive compensation (unrelated to salary bases). A sustainable company should prevent the events or phenomena mentioned above and beyond, avoiding behaviors affected by moral hazards. However, sustainability is an open concept with many interpretations and context-specific understanding. According to a pioneering perspective, sustainability is composed of the following three interconnected pillars: economic, environmental, and social. This framework is substantially based on balancing trade-offs among theoretically equally desirable goals concerning these three dimensions, although uses are different. In [1], the authors described these three dimensions in detail. While the environmental dimension

is related to the "continued productivity and functioning of ecosystems" as well as the "protection of genetic resources and the conservation of biological diversity" and the social dimension is related to "continued satisfaction of basic human needs" of individuals, the economic dimension deals with "the limitations that a sustainable society must place on economic growth". Similarly, Ref [2–4] define corporate sustainability as the systematic efforts corporations manage to balance environmental and social with economic goals to minimize negative impacts and maximize benefits for the natural environment and community. Although the literature has widely used the three-pillar paradigm, just described, as a lens to interpret the results thanks to the Brundtland Report, Agenda 21, and the 2002 World Summit on Sustainable Development [5], this paradigm is not universal as none of the documents we have just cited reports a thorough theoretical background. However, the SDGs offer several goals and instruments that can be described considering the acronym ESG (environmental, social, governance) as the lens used by investors to investigate the extra-financial sustainability of the companies belonging to their investment portfolios, and also to investigate how this sustainability is exposed to important and heterogeneous industry effects [6].

Another relevant issue concerns the nature and types of information relating to sustainability. It is known that this information is kept private, except for the companies considered by the EU Directive on non-financial reporting (EU Directive n.214/95), and, therefore, its disclosure to the public is a voluntary corporate decision. For this reason, even though most companies are interested in the evolution of ESG factors and want to give investors extra-financial information, there are different reporting approaches because of the absence of standardized rules. This circumstance has important implications on the effectiveness of this information for academic as well as business or investment purposes [7].

All in all, while we acknowledge the role of all the possible dimensions of sustainability, we focus on economic sustainability as the target to guarantee long-term growth and survival. The ESG factors are taken into consideration to measure the role played by each dimension of sustainability regarding corporate social responsibility (CSR). We collect all the sustainability data from Refinitiv Eikon (Refinitiv).

When we turn to innovation, it is well-known that this is a multifaceted phenomenon, not easy to quantify. In the economics and management of innovation literature, different proxies are considered to measure its various dimensions. There is a wide-ranging consensus that no matter how it is measured, it tends to be positive for growth, productivity, and employment [8]. In general, the ratio of research and development (R&D) expenditures to total assets (or revenues) represents the most renowned proxy for innovation, both at the macro and the micro-level. Companies' propensity to innovate generally reflects their long-term perspective [9,10]. However, R&D expenditures could not sufficiently represent the degree of a firm's innovation; a high R&D intensity is a necessary condition, but not sufficient, to be successfully innovative, inasmuch as R&D investments might not improve future performance and might not translate into valuable innovation. Indeed, Ref. [11] discuss how investors initially under-react to increases in R&D intensity. This mispricing phenomenon calls for greater attention to R&D evaluation and perception. Therefore, to quantify all the effects generated by innovation, we take into account the following two types of information: on the one hand, the information about R&D intensity and, on the other hand, the information related to growth opportunities perception (proxied by the market-to-book ratio) implied in the stock market. For this reason, our proxy for innovation is the market-perceived innovation (*MPI*) and coincides with the product between R&D intensity and the market-to-book ratio. In our opinion, the positive interaction between the last two variables represents a necessary and sufficient condition for firm innovation to be effective on economic performance and sustainability. Our choice is well-rooted, in large part, in the literature that positively links the R&D spending and its market value or market-to-book ratio or Tobin's Q [12–15].

This paper combines different literature by primarily investigating the relationship between highly valuable innovation, our *MPI*, and economic sustainability. We also in-

vestigate the relationship between ESG performance and the (sustainable) economic performance (e.g., [16,17]). To test our interest relationship, we perform a fixed-effect panel analysis, using the sample of the most innovative worldwide companies selected by the European Commission's Joint Research Centre (EC JRC), covering the 2013–2017 time span.

In light of the above, this work complements literature in several ways. First, we offer empirical evidence for worldwide companies' MPI and their economic sustainability. More precisely, we show a positive association between economic sustainability and our firm innovation measure, confirming that a virtuous paradigm is achievable.

Second, we show that the traditional pillars of sustainability might not be mutually independent. Economic sustainability is positively affected by the social dimension and is less related to the environmental and corporate governance dimensions. These last results can be very useful for policymakers to identify reform priorities depending on the desirable targets.

We also show that results are confirmed even when we control for context-specific conditions, both at the geographical and industrial levels.

The paper is organized as follows: Section 2 provides an overview of the main arguments used in prior literature, Section 3 defines our sample, methodology, and measures, while Section 4 discusses the main results of our empirical analysis. Section 5 provides concluding remarks.

## 2. Literature and Hypothesis Development

Over the years, the majority of the authors have focused their attention on the relationship between different ESG factors and financial performance [18], sometimes finding positive relationships (see, for example, [19–21]). Other authors [22,23] have empirically shown a positive relationship between corporate sustainability and corporate financial performance. So, while we consider the ESG/CSR approach as a dimension that needs to be considered when economic sustainability—our focus—is analyzed, we aim at better understanding the role played by innovation.

First of all, when innovation is considered, the literature is extensive due to its multifaceted nature and potential effects. R&D expenditures and R&D intensity are the most well-known proxies able to grasp the innovative investments of companies. While these measures are reliable in terms of effort, there is no guarantee that innovative efforts are translated into innovative output relevant to the company and the economy [24,25].

Some authors investigate the relationship between innovation and economic performance. Refs. [26,27] highlight that all firms need a strategy to integrate sustainable development and innovation. In addition, Ref. [28] state that long-term firm success derives from balancing both the competing and complementary interests of stakeholder segments, including community and environment, to increase the likelihood of sustainable competitive advantage. Focusing on green innovation, Ref. [29] emphasize that the performances of green product innovation and green process innovation are positively related to the corporate competitive advantage, i.e., businesses can benefit from investments in green product and process innovations. In this research line, Ref. [30] highlight how corporate economic sustainability is positively associated with growth and, therefore, with company innovation. This evidence is confirmed by Refs. [31,32].

Other works document that R&D intensity might significantly improve firms' financial performance [33,34]. Firms tend to perform better when investing in R&D to introduce a new product or modify a production process.

However, from the market perspective, R&D could not sufficiently represent the degree of a firm's innovation; a high R&D intensity is a necessary condition, but it might not be sufficient to be effectively innovative, inasmuch as R&D investments might not translate into MPI. The works cited above do not consider the meaning of innovation and how it combines with companies' growth opportunities, impacting sustainability. In other words, R&D expenses may not necessarily represent final successful innovation and

may be a necessary but not sufficient condition, unless such investments generate growth opportunities that some investors, or the market, can recognize.

Refs. [12–14] show that innovation–growth opportunities interaction can be a cause—effect logic or a one-way relationship, and can be conditioned by firm characteristics (size, market share, industry). Refs. [35,36] analyze the UK market and the US market together with the Japanese one, and confirm moderating effects through industry and size. Ref. [37] confirms that R&D expenses tend to influence growth opportunities as follows: as R&D intensities increase, the market-to-book ratio increases. In [38], the authors link growth opportunities also with financial leverage. Recently, in [39], the authors show how innovation, especially in green technologies, positively affects Tobin's Q in the automotive industry. In [40], the authors suggest possible correlations between R&D and growth depending on specific firm characteristics. From this point of view, we think that high-intensity R&D firms with high market-to-book ratios represent firms that invest a lot and have high expected growth, implied in market prices. The function of the market (or a group of investors—for example, venture capitalists) is crucial; indeed, the market recognizes the value of innovation and, therefore, of expected growth. This role is central not only in supporting investments but also in carrying out a function of selection among the various firms attempting to innovate.

This reasoning leads us to think that, for sustainability, innovation can be captured by a combination or interaction between R&D intensity and business growth opportunities (MPI). Indeed, in [11], the authors show how investors initially tend to under-react to increments of R&D intensity. This mispricing requires greater attention to R&D assessment.

Starting from this research avenue, we decided to use a combination of R&D intensity and the market-to-book ratio as a proxy for firm MPI to bridge a significant gap in the literature concerning whether the companies that systematically manage to innovate are also sustainable. Figure 1 represents our main hypothesis where, in a theoretical scatter, we hypothesize that virtuous interaction between R&D intensity and the market-to-book ratio (MtBV) is the expected growth market dimension. The efficient combination (squared points on the diagonal) positively affects economic sustainability. When both are high (H-H), we might expect a positive effect on economic sustainability. When both are low (L-L), in a coherent setting, the low innovative effort is accompanied by a low MtBV and limited expected effect on economic sustainability. In the remaining quadrants (L-H) and (H-L), the two dimensions are not moving harmoniously nor efficiently, as follows: in (L-H), the low innovative effort seems to be excessively rewarded by the market; in (H-L), the innovative effort does not receive a premium by the market. Therefore, we want to investigate the relationship between MPI and economic sustainability as our evidence could confirm the hints already emerged in the literature and offer new insights.

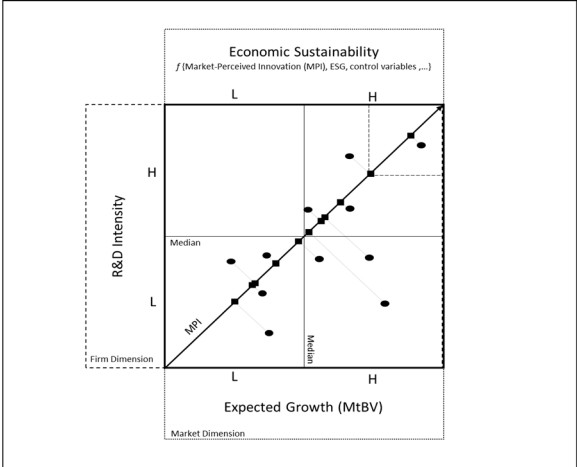

**Figure 1.** Theoretical scatter for the relationship between economic sustainability and MPI, combining R&D intensity and the expected growth opportunities of a firm.

Our first hypothesis is, therefore, as follows:

**Hypothesis 1 (H1).** *The relationship between MPI and firm economic sustainability is positive.*

An additional factor of interest is represented by the complex relationships among individual pillars of sustainability—ESG factors—as they influence the ways firms achieve sustainability. Identifying the characteristics of the causalities among sustainability pillars can be very important from a policy point of view as most governments have wanted to improve their sustainable development since the Brundtland report [41]. For governments, the best solution would be to design sustainability as an integrative process. For example, in [42], the authors suggest that CSR is seemingly recession-proof as CSR's commitment is conservative and guarantees improvements in the safe area of efficiency. Therefore, according to the Brundtland report (1987:54), "economic and social development should be mutually reinforcing." Moreover, as, according to [43], the economic pillar of sustainability reflects the need to balance the costs and benefits of economic activity, respecting the environmental constraints, economic progress should not damage intergenerational equity. Therefore, resources should be used to guarantee their regenerative ability.

For this reason, we investigate the relationship between single ESG pillars of sustainability (environmental, social, and governance) and economic sustainability formulating our second group of hypotheses, as follows:

**Hypothesis 2a (H2a).** *The relationship between the firm environmental dimension and the firm economic sustainability is positive.*

**Hypothesis 2b (H2b).** *The relationship between the firm social dimension and the firm economic sustainability is positive.*

**Hypothesis 2c (H2c).** *The relationship between the firm corporate governance dimension and the firm economic sustainability is positive.*

In addition, as some authors [44,45] show that sustainable corporate performance might be conditional on context-specific conditions, we can also assume that our relationships of interest, both that between MPI and economic sustainability and those between the ESG factors and economic sustainability, are affected if firms have headquarters in some macro-regions or belong to a specific industry. From the geographical side, macro-regional institutional settings might influence the amount of investments in innovation and the premium paid by the markets [46]. Instead, from the sectoral specialization side, to select the most important sustainability dimensions for policy reforms in each industry, Ref. [6] shows that for consumer non-cyclical, healthcare, industrial, technology, telecommunication, and utility industries, the most important role is played by social performance. On the contrary, environmental performance is the most critical pillar for basic material, consumer cyclical, and financial industries, while for the energy industry, economic performance is the most critical pillar. The governance performance, however, is not significant in any industry.

Therefore, we formulate our last hypotheses, as follows:

**Hypothesis 3a (H3a).** *The relationship between MPI/ESG and economic sustainability depends on the firm's macro-region.*

**Hypothesis 3b (H3b).** *The relationship between MPI/ESG and economic sustainability depends on the firm's industry.*

## 3. Data and Methodology

To study our main relationships of interest, we use a company-level dataset, obtained merging two data sources as follows: EC JRC [providing the list of the 2500 most innovative companies in the world (ranked for R&D expenses) and Refinitiv providing ESG information in terms of scores concerning environmental, social, governance and economic sustainability dimensions. The EC JRC elaborates an annual edition of the EU Industrial R&D Investment Scoreboard comprising the 500 companies investing the largest sums in R&D in the world. Overall, they represent approximately 90% of the world's business-funded R&D.] Hence, using the EC JRC database, we selected the champions of innovation. Therefore, we focus on the most innovative companies worldwide, as their innovative effort is likely to be more recognizable and well-priced by the market. In our MPI perspective, we aim at focusing on the (H-H) quadrant in Figure 1 [Some countries, such as US and Japan, are overrepresented (in 2020 [47], US companies are more than the 35% and Japanese approx. 15%). Moreover, the innovative success of a small number of top R&D investors might determine business stealing and increasing concentration dynamics at the sectoral level. However, our aim is to test whether high innovation effort well-recognized by the market positively impacts economic sustainability. This approach can be replicated for less innovative companies, but data on R&D, disclosure of information and market perception might be more biased than in the case of the champions.]. In addition, on the other hand, Refinitiv provides ESG scores for more than 4000 firms by evaluating all companies listed in primary market indexes such as CAC 40, DAX, MSCI World, NASDAQ 100, S&P 500, STOXX 600, and others [48]. This process strongly reduces the risk of selection bias since each firm is analyzed despite its specific ESG and communication strategy. In the same Refinitiv database, we also collect the most common accounting information except for R&D expenses (i.e., total sales, total assets, and long-term debt). Our final sample is composed of the 2500 most innovative companies indicated by the EC JRC with ESG information available in Refinitiv over the 2013-17 time-span, which is 909 worldwide firms.

We estimate the panel data using model (1)

$$y_{it} = [di \; X_{it}] \begin{bmatrix} \alpha \\ \beta \end{bmatrix} + \varepsilon_{it} \quad I = 1, \dots, N; \; t = 1, \dots, T \tag{1}$$

where $y_{it}$ is our dependent variable (economic sustainability) for firm i at time t, $d_i$ is a dummy variable indicating the ith unit, $X_{it}$ is a vector of covariates, $\alpha$ and $\beta$ are vectors of coefficients, $\varepsilon_{it}$ indicates a zero-mean idiosyncratic stochastic error term. Our model is referred to as the least squares dummy variable (LSDV) model and allows solving a great shortcoming of the fixed effects approach (FE) as follows: any time-invariant variables in $X_{it}$ will mimic the individual specific constant term [49]. As N, the number of our firms is small enough, the model can be estimated by ordinary least squares with K regressors in X and N columns in D, as a multiple regression with K + N parameters. The standard errors of our estimators are robust. Our choice of using a fixed panel model [should capture potential firm or time characteristics; for example, we expect that economic sustainability is influenced by time effects, given the growing media interest (or pressure) in the last decade. Our choice of using the fixed effects estimator is properly justified as we apply a Hausman Test to all our models, according to which we reject H0 of preferring the random effects estimator.

To measure the effect of covariates on economic sustainability, we regress *Ecnscore* on the covariates. Our variable of interest, *MPI*, is the product between R&D intensity (R&D expenditures on total assets) and the market-to-book ratio (*MtBV*). Instead of considering the two single variables' separate effects, we focus on their interaction getting ideas from the studies that show a positive relationship between R&D intensity and the market-to-book ratio (see Section 2, Hypothesis 1). The combination of these two variables expresses R&D intensity amplified/multiplied or discounted by the market-to-book ratio (the growth opportunities implied in the market valuation) and, therefore, MPI. The second variable

of interest is the aggregate ESG score that allows us to investigate how the sustainability pillars are correlated among them. We also investigate the effects of specific ESG factors (environmental score, social score, corporate governance score) or dimensions on economic sustainability. The idea of looking for more detailed information is also motivated by the need to confirm or refuse the estimates obtained using the aggregate score (Hypothesis 2).

We also use other variables to control the effects due to size (proxied by the natural logarithm of revenues), profitability (proxied by ROA), and financial leverage (proxied by the ratio of long-term debt to total assets).

While Table 1 provides the list of each variable used in our analysis and its corresponding definition, Table 2 shows both descriptive statistics and the matrix correlation.

**Table 1.** Definitions and data sources for every variable of our analyses.

| Variables | Definitions | Data Source |
|---|---|---|
| **Dependent variable** | | |
| *Ecnscore* | The economic pillar measures a company's capacity to generate sustainable growth and a high return on investment through the efficient use of all its resources. It reflects a company's overall financial health and its ability to generate long-term shareholder value through its use of best management practices. | Refinitiv |
| **Independent variable** | | |
| *MPI* | The product of annual R&D intensity defined as the ratio of R&D expenses to total assets multiplied by the market-to-book ratio. | EC JRC (1° Factor); Refinitiv (2° Factor) |
| *Esgscore* | The simple average of *Envscore, Socscore*, and *Cgvscore*. | Author's calculation |
| *Envscore* | The environmental pillar measures a company's impact on living and non-living natural systems, including the air, land, and water, as well as complete ecosystems. It reflects how well a company uses best management practices to avoid environmental risks and capitalize on environmental opportunities to generate long-term shareholder value. | Refinitiv |
| *Socscore* | The social pillar measures a company's capacity to generate trust and loyalty with its workforce, customers, and society through its use of best management practices. It is a reflection of the company's reputation and the health of its license to operate, which are key factors in determining its ability to generate long-term shareholder value. | Refinitiv |
| *Cgvscore* | The corporate governance pillar measures a company's systems and processes, which ensures that its board members and executives act in the best interests of its long-term shareholders. It reflects a company's capacity through its use of best management practices to direct and control its rights and responsibilities through the creation of incentives as well as checks and balances to generate long-term shareholder value. | Refinitiv |
| *China* | Dummy variable equals to 1 if the firm country is China, 0 otherwise. | Refinitiv |
| *EU* | Dummy variable equals to 1 if the firm country belongs to European Union, 0 otherwise. | Refinitiv |
| *Japan* | Dummy variable equals to 1 if the firm country is Japan, 0 otherwise. | Refinitiv |
| *Northamerica* | Dummy variable equals to 1 if the firm country is the US or Canada, 0 otherwise. | Refinitiv |
| *Tech* | Dummy variable equals 1 if the firm industry is the technological, 0 otherwise. | Orbis (Bureau van Dijk) |
| **Control variable** | | |
| *Size* | Natural logarithm of the total sales. | Refinitiv |
| *Profitability* | Ratio of EBIT to total assets. | Refinitiv |
| *Leverage* | Ratio of long-term debt to total assets. | Refinitiv |

**Table 2.** Descriptive statistics.

| | n. obs | Firms | Mean | st. dev. | min | MAX | 1 | 2 | 3 | 4 | 5 | 6 | 7 | 8 | 9 |
|---|---|---|---|---|---|---|---|---|---|---|---|---|---|---|---|
| *Ecnscore* | 4246 | 909 | 68.66 | 24.77 | 2.54 | 98.29 | 1 | | | | | | | | |
| *MPI* | 4246 | 909 | 13.38 | 21.25 | 0.11 | 121.83 | −0.0391 | 1 | | | | | | | |
| *Esgscore* | 4246 | 909 | 65.79 | 21.56 | 5.27 | 96.68 | 0.6308 * | 0.0166 | 1 | | | | | | |
| *Envscore* | 4246 | 909 | 74.74 | 26.87 | 8.63 | 95.59 | 0.5381 * | −0.191 * | 0.7659 * | 1 | | | | | |
| *Socscore* | 4246 | 909 | 71.14 | 26.00 | 4.38 | 97.55 | 0.6698 * | −0.0884 * | 0.8594 * | 0.7866 * | 1 | | | | |
| *Cgvscore* | 4246 | 909 | 51.50 | 33.58 | 1.11 | 97.82 | 0.2657 * | 0.2533 * | 0.6479 * | 0.0662 * | 0.2517 * | 1 | | | |
| *Size* | 4246 | 909 | 15.54 | 1.20 | 13.5 | 17.88 | 0.3541 * | −0.2301 * | 0.3954 * | 0.4371 * | 0.4231 * | 0.0843 * | 1 | | |
| *Profitability* | 4246 | 909 | 6.019 | 4.90 | −2.77 | 16.69 | 0.1695 * | 0.2780 * | 0.1095 * | −0.0450 * | 0.0268 | 0.2262 * | −0.0954 * | 1 | |
| *Leverage* | 4246 | 909 | 17.41 | 12.90 | 0 | 44.82 | 0.0796 * | −0.0133 | 0.2649 * | 0.0820 * | 0.1415 * | 0.3350 * | 0.1889 * | −0.1220 * | 1 |

* for *p*-value < 0.01.

While the mean of our dependent variable (*Ecnscore*) is 68.66, the mean of our main independent variable (*MPI*) is 13.38. Our sample firms, which are characterized by the highest R&D expenses, seem to be more interested in the environmental and social dimensions than the corporate governance one, as follows: the mean values of the first two dimensions are 74.74 and 71.14, respectively, while the mean of the last dimension is 51.50. The leverage ratio is low, ranging from 0 to 44.82%, with a mean value of 17.41%.

Table 3 provides a sample breakdown by year, industry, and country. In general, we can say that all our main variables (innovation and ESG variables) increase over time during our analysis period. If we focus on industries, the real estate industry is characterized by the highest mean values for *Ecnscore* (84.06), *Esgscore* (90.55), *Envscore* (92.60), *Socscore* (93.62), and *Cgvscore* (85.42). On the contrary, the travel and leisure industry records the lowest values in terms of *Esgscore* (44.78), *Envscore* (36.57), and *Socscore* (38.50). If we focus on countries, we identify no cluster.

**Table 3.** Sample breakdown by year, industry, and country.

| Panel A | | | | | | | | | | | |
|---|---|---|---|---|---|---|---|---|---|---|---|
| Year | *Ecnscore* | | *MPI* | | *Esgscore* | | *Envscore* | | *Socscore* | | *Cgvscore* |
| | n | mean | n | Mean | n | mean | n | mean | n | mean | N | mean |
| 2013 | 879 | 58.10 | 879 | 12.87 | 879 | 60.24 | 879 | 68.39 | 879 | 62.49 | 879 | 49.85 |
| 2014 | 871 | 61.62 | 871 | 13.57 | 871 | 61.71 | 871 | 70.21 | 871 | 64.85 | 871 | 50.06 |
| 2015 | 848 | 69.67 | 848 | 13.19 | 848 | 66.37 | 848 | 75.31 | 848 | 72.21 | 848 | 51.59 |
| 2016 | 839 | 76.35 | 839 | 13.05 | 839 | 69.58 | 839 | 79.13 | 839 | 77.16 | 839 | 52.46 |
| 2017 | 809 | 78.67 | 809 | 14.29 | 809 | 71.68 | 809 | 81.34 | 809 | 79.95 | 809 | 53.73 |
| Total | 4246 | | 446 | | 4246 | | 4246 | | 4246 | | 4246 | |

| Panel B | | | | | | | | | | | |
|---|---|---|---|---|---|---|---|---|---|---|---|
| Industry | *Ecnscore* | | *MPI* | | *Esgscore* | | *Envscore* | | *Socscore* | | *Cgvscore* |
| | n | mean | n | mean | n | mean | n | mean | n | mean | N | mean |
| Automobile and Parts | 260 | 65.21 | 260 | 9.29 | 260 | 63.74 | 260 | 82.69 | 260 | 68.08 | 260 | 40.46 |
| Basic Resource | 169 | 70.80 | 169 | 1.06 | 169 | 68.89 | 169 | 85.20 | 169 | 79.07 | 169 | 42.40 |
| Chemicals | 325 | 77.78 | 325 | 6.00 | 325 | 71.36 | 325 | 87.56 | 325 | 82.86 | 325 | 43.66 |
| Constr e Mat | 138 | 75.79 | 138 | 5.87 | 138 | 67.43 | 138 | 84.61 | 138 | 76.28 | 138 | 41.41 |
| Consum Prod e Serv | 321 | 68.30 | 321 | 10.80 | 321 | 61.63 | 321 | 71.59 | 321 | 67.95 | 321 | 45.36 |
| Drug e Grocery Store | 53 | 81.96 | 53 | 11.50 | 53 | 81.17 | 53 | 88.15 | 53 | 87.94 | 53 | 67.41 |
| Energy | 152 | 76.43 | 152 | 1.78 | 152 | 75.30 | 152 | 80.58 | 152 | 79.39 | 152 | 65.93 |
| Financial service | 24 | 56.94 | 24 | 5.09 | 24 | 45.99 | 24 | 47.12 | 24 | 41.06 | 24 | 49.78 |
| Bev Food Tob | 200 | 74.09 | 200 | 4.14 | 200 | 67.80 | 200 | 74.77 | 200 | 75.61 | 200 | 53.01 |
| Healt Care | 471 | 66.90 | 471 | 29.85 | 471 | 63.22 | 471 | 61.03 | 471 | 68.23 | 471 | 60.41 |
| Ind Goods and Service | 969 | 65.79 | 969 | 9.40 | 969 | 65.44 | 969 | 75.32 | 969 | 67.86 | 969 | 53.15 |
| Media | 25 | 51.18 | 25 | 27.00 | 25 | 56.39 | 25 | 58.91 | 25 | 51.42 | 25 | 58.82 |
| Real Estate | 5 | 84.06 | 5 | 0.58 | 5 | 90.55 | 5 | 92.60 | 5 | 93.62 | 5 | 85.42 |
| Retailer | 15 | 55.41 | 15 | 41.10 | 15 | 54.95 | 15 | 53.03 | 15 | 65.58 | 15 | 46.25 |
| Technology | 835 | 66.28 | 835 | 23.00 | 835 | 64.29 | 835 | 71.65 | 835 | 69.43 | 835 | 51.79 |
| Telecomm | 150 | 71.36 | 150 | 12.55 | 150 | 71.97 | 150 | 73.95 | 150 | 75.80 | 150 | 66.16 |
| Travel & Leisure | 25 | 51.45 | 25 | 23.23 | 25 | 44.78 | 25 | 36.57 | 25 | 38.50 | 25 | 59.28 |
| Utility | 109 | 70.40 | 109 | 0.32 | 109 | 63.96 | 109 | 82.75 | 109 | 73.07 | 109 | 36.06 |
| Total | 4246 | | 4246 | | 4246 | | 4246 | | 4246 | | 4246 | |

| Panel C | | | | | | | | | | | |
|---|---|---|---|---|---|---|---|---|---|---|---|
| Country | *Ecnscore* | | *MPI* | | *Esgscore* | | *Envscore* | | *Socscore* | | *Cgvscore* |
| | n | mean | n | mean | n | mean | n | mean | n | mean | N | mean |
| Australia | 45 | 73.75 | 45 | 30.98 | 45 | 71.07 | 45 | 62.95 | 45 | 67.39 | 45 | 82.86 |
| Austria | 21 | 77.02 | 21 | 2.03 | 21 | 71.68 | 21 | 79.17 | 21 | 79.74 | 21 | 56.13 |
| Belgium | 25 | 79.68 | 25 | 13.03 | 25 | 80.84 | 25 | 86.76 | 25 | 85.84 | 25 | 69.93 |
| Bermuda | 32 | 35.57 | 32 | 5.65 | 32 | 42.25 | 32 | 40.61 | 32 | 31.34 | 32 | 54.79 |
| Brazil | 17 | 67.69 | 17 | 20.90 | 17 | 58.11 | 17 | 59.71 | 17 | 73.16 | 17 | 41.47 |
| Canada | 30 | 71.44 | 30 | 12.97 | 30 | 62.80 | 30 | 55.49 | 30 | 54.35 | 30 | 78.55 |
| Cayman Islands | 58 | 39.72 | 58 | 6.35 | 58 | 46.79 | 58 | 50.73 | 58 | 44.73 | 58 | 44.91 |
| China | 54 | 37.02 | 54 | 2.56 | 54 | 41.11 | 54 | 44.64 | 54 | 37.84 | 54 | 40.85 |
| Curaçao | 5 | 84.83 | 5 | 4.17 | 5 | 87.68 | 5 | 83.68 | 5 | 87.85 | 5 | 91.50 |
| Czech Republic | 1 | 72.85 | 1 | 0.19 | 1 | 68.85 | 1 | 89.51 | 1 | 85.67 | 1 | 31.38 |

**Table 3.** *Cont.*

| | | | | | | | | | | | | |
|---|---|---|---|---|---|---|---|---|---|---|---|---|
| Denmark | 42 | 78.82 | 42 | 26.34 | 42 | 78.60 | 42 | 85.12 | 42 | 78.48 | 42 | 72.19 |
| Germany | 194 | 75.81 | 194 | 6.91 | 194 | 70.92 | 194 | 83.89 | 194 | 82.68 | 194 | 46.18 |
| Finland | 73 | 85.73 | 73 | 7.85 | 73 | 81.94 | 73 | 91.94 | 73 | 87.07 | 73 | 66.82 |
| France | 167 | 80.35 | 167 | 7.09 | 167 | 84.04 | 167 | 91.72 | 167 | 90.19 | 167 | 70.21 |
| Hong Kong | 25 | 87.18 | 25 | 4.74 | 25 | 71.27 | 25 | 63.55 | 25 | 76.25 | 25 | 74.02 |
| Hungary | 5 | 50.25 | 5 | 8.30 | 5 | 42.10 | 5 | 36.84 | 5 | 58.82 | 5 | 30.65 |
| India | 82 | 77.93 | 82 | 7.00 | 82 | 70.95 | 82 | 80.31 | 82 | 78.47 | 82 | 54.07 |
| Ireland | 26 | 78.59 | 26 | 22.71 | 26 | 84.12 | 26 | 85.83 | 26 | 79.49 | 26 | 87.04 |
| Israel | 25 | 54.73 | 25 | 9.85 | 25 | 48.52 | 25 | 50.67 | 25 | 47.61 | 25 | 47.28 |
| Italy | 23 | 90.60 | 23 | 3.01 | 23 | 89.22 | 23 | 92.22 | 23 | 91.55 | 23 | 83.89 |
| Japan | 1049 | 67.58 | 1049 | 5.87 | 1049 | 54.46 | 1049 | 82.03 | 1049 | 70.03 | 1049 | 11.33 |
| Luxembourg | 10 | 78.44 | 10 | 0.55 | 10 | 59.77 | 10 | 59.27 | 10 | 64.24 | 10 | 55.80 |
| Malaysia | 8 | 85.47 | 8 | 0.32 | 8 | 85.38 | 8 | 89.50 | 8 | 82.23 | 8 | 84.40 |
| Netherlands | 45 | 81.81 | 45 | 25.17 | 45 | 83.38 | 45 | 82.15 | 45 | 85.42 | 45 | 82.57 |
| New Zealand | 5 | 57.51 | 5 | 73.49 | 5 | 55.58 | 5 | 40.32 | 5 | 56.97 | 5 | 69.45 |
| Norway | 27 | 79.86 | 27 | 2.56 | 27 | 78.25 | 27 | 84.00 | 27 | 81.50 | 27 | 69.26 |
| Saudi Arabia | 5 | 94.54 | 5 | 0.92 | 5 | 68.67 | 5 | 93.45 | 5 | 88.67 | 5 | 23.88 |
| Singapore | 10 | 71.66 | 10 | 4.24 | 10 | 60.87 | 10 | 54.63 | 10 | 56.47 | 10 | 71.52 |
| South Africa | 5 | 80.90 | 5 | 0.82 | 5 | 93.07 | 5 | 93.55 | 5 | 95.08 | 5 | 90.58 |
| South Korea | 156 | 63.03 | 156 | 5.12 | 156 | 51.41 | 156 | 72.64 | 156 | 67.81 | 156 | 13.78 |
| Spain | 19 | 85.81 | 19 | 14.35 | 19 | 81.63 | 19 | 91.87 | 19 | 95.27 | 19 | 57.75 |
| Sweden | 56 | 76.28 | 56 | 6.99 | 56 | 77.65 | 56 | 88.56 | 56 | 86.93 | 56 | 57.45 |
| Switzerland | 156 | 72.85 | 156 | 18.66 | 156 | 69.46 | 156 | 74.60 | 156 | 74.23 | 156 | 59.54 |
| Taiwan | 264 | 54.94 | 264 | 8.73 | 264 | 53.67 | 264 | 72.36 | 264 | 65.84 | 264 | 22.81 |
| Turkey | 17 | 77.94 | 17 | 3.77 | 17 | 67.52 | 17 | 92.04 | 17 | 76.70 | 17 | 33.81 |
| United Kingdom | 213 | 72.90 | 213 | 16.12 | 213 | 76.96 | 213 | 74.75 | 213 | 74.07 | 213 | 82.06 |
| United States | 1236 | 67.62 | 1236 | 23.93 | 1236 | 71.93 | 1236 | 66.69 | 1236 | 68.23 | 1236 | 80.89 |
| Total | 4231 | | 4231 | | 4231 | | 4231 | | 4231 | | 4231 | |

## 4. Empirical Results

To study our relationships of interest, we regress economic sustainability on our innovation proxy (*MPI*), ESG scores, and several control variables (every regression signals the existence of fixed effects as confirmed by a Hausman test). *MPI* is defined as the product between R&D intensity and the market-to-book ratio; it measures the firm R&D commitment (intensity), amplified/discounted by market growth perceptions. Table 4 highlights the results of our two baseline regression models. As this relationship is positive and statistically significant in both models (1) and (2), MPI confirms to be a determinant for sustainable economic performance, and hence we do not reject Hypothesis 1. The magnitude of the estimated coefficient suggests an expected change of about 5% (0.0492 in model 1 and 0.0478 in model 2) in economic sustainability for a one-point change in MPI. The standardized coefficient is equal to about 0.0422, that is, the estimated coefficient (0.0492) multiplied by the ratio between the standard deviations of the covariate (21.25) and the dependent variable (24.77). In our opinion, the relevance (significance) of *MPI* is attributable to the combination of the different information content of the two factors, as follows: the R&D intensities represent the internal financial efforts (or commitment) of the company, which are amplified/multiplied or discounted by the growth opportunities implied in the valuation of the market investors (or any specific investors such as venture capitalists). If the market-to-book ratio (growth proxy) is higher (lower) than one, the firm financial effort is amplified (discounted). In summary, MPI identifies a condition that is not only necessary but also sufficient to capture the calibration process of the internal financial resources to the external opportunities. The higher the firm's managerial ability to combine internal resources with external opportunities, the higher the economic sustainability. The firms try to generate sustainable growth and efficiently use all their resources. In other words, more innovation-oriented firms tend to be more committed to generating long-term shareholder value through better management practices.

**Table 4.** Baseline models. The dependent variable is economic sustainability (*Ecnscore*). The models are fixed effects panel regression models (robust standard errors). Model 1 focuses on the aggregate ESG score, while model 2 focuses on the single dimensions of ESG. For every independent variable, we report the coefficient, the standard error, and the statistical significance: *** for *p*-value < 0.01, ** for *p* < 0.05 and * for *p* < 0.1. Data are collected from 2013 to 2017. Data source: EC JRC and Refinitiv.

| | Model 1 | | | Model 2 | | |
|---|---|---|---|---|---|---|
| | coeff. | se | Sig | coeff. | se | sig |
| *constant* | −32.4684 | 19.3043 | * | −24.0527 | 19.2630 | |
| *MPI* | 0.0492 | 0.0248 | ** | 0.0478 | 0.0247 | * |
| *Esgscore* | 0.5085 | 0.0322 | *** | | | |
| *Envscore* | | | | 0.0569 | 0.0286 | ** |
| *Socscore* | | | | 0.3314 | 0.0266 | *** |
| *Cgvscore* | | | | 0.0752 | 0.0287 | *** |
| *Size* | 3.8767 | 1.0890 | *** | 3.6379 | 1.0826 | *** |
| *Profitability* | 0.7288 | 0.0685 | *** | 0.7337 | 0.0682 | *** |
| *Leverage* | −0.2926 | 0.0380 | *** | −0.2924 | 0.0378 | *** |
| Firm fixed effects | Yes | | | Yes | | |
| Year fixed effects | Yes | | | Yes | | |
| N | 4246 | | | 4246 | | |
| Adjusted $R^2$ | 0.8124 | | | 0.8150 | | |

Not surprisingly, our models also confirm a positive and statistically significant relationship between economic sustainability and ESG scores independently if we consider the aggregate score or its main subdimensions, that is, environmental, social, and governance. These findings suggest Hypothesis 2 (a, b, c) should not be rejected. The change of one point in the ESG aggregate score is associated with an impact of 0.5085, equivalent to a standardized coefficient of about 0.4426 (0.5085 × 21.56/24.77). Therefore, the ESG score turns out to be the major determinant of economic sustainability among the variables under analysis. Even when investigating the individual pillars, the strong relevance does not change, although the impact is different. In fact, for the change of a point of the social pillar, the expected change in economic sustainability equals 0.3314, and, therefore, an expected impact of about 0.3479 (0.3314 × 26/24.77), the highest of all the variables. The fact that the other two pillars show more limited influences is attributable to the characteristics of the companies in the sample; for example, the environmental pillar shows a coefficient (expected change on economic sustainability) of 0.0569, standardized equal to about 0.0617 (0.0569 × 26.87/24.77) and such a weak influence could depend on the fact that companies more innovation-oriented (such as those of the sample) are less exposed to environmental risks and, in fact, the sample mean of the pillar (74.74) is higher than the others. The governance pillar shows an impact (standardized coefficient) almost double compared to the environmental one and about 0.1020 (0.0752 × 33.58/24.77) but just one-third compared to the social one, however, signaling/confirming that the issues of governance are not negligible for economic sustainability. These results lead us to think that companies should pay particular attention to the social pillar, which negatively affects economic sustainability through, for example, reputational risks, poor compliance in terms of human rights, safety at work and so on.

If we focus on the control variables, the corresponding coefficients confirm the expectations and relationships in the literature, according to [12,13,44]. Specifically, the size factor (proxied by the natural logarithm of revenues) shows a positive and significant association with economic sustainability in all our models; the estimated coefficient equal to 3.8767 in model 1 (and 3.6379 in model 2) expresses the expected change of the economic sustainability score for a variation of one unit of the size factor, which in turn, therefore, corresponds to a change of approximately USD 2.72 billion of sales. This evidence suggests and confirms the size as a significant, positive and determining factor; indeed, the standardized coefficient shows an impact of about 0.1878 (3.8767 × 1.2/24.77) of the size

on economic sustainability, an impact certainly more relevant than other variables. The profitability factor also exhibits a positive and significant contribution, showing plausibly that greater financial resources (typically of large companies with high profits) correspond to stronger economic sustainability. For a change of one percentage point of profitability (measured with the ratio of EBIT on total assets), model 1 estimates a positive change in the economic sustainability score equal to 0.7288 (substantially similar in model 2, with a score of 0.7337), which corresponds to an impact of about 0.1442 ($0.7288 \times 4.9/24.77$) similar to the size factor. Otherwise, a change of one percentage point in financial leverage, which incorporates the effects of financial distress, highlights an expected change in economic sustainability equal to $-0.2926$ in model 1 (completely similar in model 2), which corresponds to a (negative) impact of approximately $-0.1524$ ($-0.2926 \times 12.9/24.77$).

To investigate the robustness or stability of our relationships of interest in the basic regressions, we test the impact of country and industry effects (Hypothesis 3) in Tables 5 and 6, respectively. It is evident how the estimated coefficients of our variables of interest (MPI and ESG score) and control variables are substantially similar to those previously exposed in models 1 and 2, confirming stable relationships that suggest a positive and constant pattern between economic sustainability, MPI and ESG factors (and control variables). However, specific differences in geographic or sector terms provide for different responses on economic sustainability (different coefficients of the country or industry dummy variables). Therefore, our dependent variable also turns out to be influenced by country and industry-specific factors, confirming Hypothesis 3 (a, b). If we focus on countries, the economic sustainability relationship is higher for the firms operating in Europe and Japan (although the latter loses significance when we control for specific ESG pillars), while it is lower for the firms operating in China. The first result confirms our expectations as European countries, except for the UK and Japan, are civil law countries characterized by a stakeholder orientation [50,51]. Corroborated literature [52,53] shows that the country's legal system influences how the society protects shareholder and debtholder interests and, hence, governance structures of firms and their decision-making processes. In other words, as the stakeholder pressure is stronger in civil law countries [54], our results confirm the idea that in these countries, managers tend to go beyond a shareholders' perspective, deciding to sacrifice the maximization of the shareholder value to satisfy the demands of the broad set of stakeholders. On the other hand, we are not surprised by the result concerning Chinese firms. China is one of the largest emerging economies, and its institutions oriented to support corporate sustainability are not strongly developed. For example, although the Central Government in China has established a specific system to protect the environment, this system is poorly implemented [55]. A similar consideration could be extended for the industry effect. Specifically, firms belonging to the technology sector tend to be associated with greater economic sustainability, according to [6,44,45].

Last but not least, each proposed regression signals the presence of firm and time fixed effects, jointly significant; this highlights that some events might have influenced our dependent variable; in particular, the economic sustainability of the companies analyzed was conditioned in the years covered by our sample (2013–2017) by the 2015 Paris Agreements on Climate Change and, therefore, also on issues of economic sustainability. This significance confirms that cooperation between nations can represent a proactive factor in encouraging companies (or, more generally, the economic system) to pay greater attention to economic sustainability issues together with innovation processes and ESG factors.

In summary, we find that the economic sustainability of high R&D-spending firms is positively and strongly associated with ESG factors (and especially with the social pillar); subsequently, the (positive) size, (negative) leverage and (positive) profitability effects impact with less intensity. We find that our MPI, which synthesizes the managerial ability of the company to combine growth opportunities with financial commitment in R&D, contributes to positively increasing economic sustainability.

**Table 5.** Country analysis. The dependent variable is economic sustainability (*Ecnscore*). The models are fixed effects panel regression models (robust standard errors). Model 3 focuses on the aggregate ESG score, while model 4 focuses on the single dimensions of ESG. Country variables are dummies equal to 1 if the firm belongs to the selected area (China, European Union, Japan, North America) and zero otherwise. For every independent variable, we report the coefficient, the standard error, and the statistical significance: *** for *p*-value < 0.01, ** for *p* < 0.05 and * for *p* < 0.1. Data are collected from 2013 to 2017. Data source: EC JRC and Refinitiv.

| | Model 3 | | | Model 4 | | |
|---|---|---|---|---|---|---|
| | coeff. | Se | sig | coeff. | Se | sig |
| *constant* | −32.7938 | 19.3096 | * | −24.3977 | 19.2666 | |
| *MPI* | 0.0489 | 0.0249 | ** | 0.0475 | 0.0247 | * |
| *Esgscore* | 0.5100 | 0.0323 | *** | | | |
| *Envscore* | | | | 0.0552 | 0.0286 | *** |
| *Socscore* | | | | 0.3336 | 0.0267 | *** |
| *Cgvscore* | | | | 0.0768 | 0.0287 | *** |
| *Size* | 3.8859 | 1.0894 | *** | 3.6450 | 1.0828 | *** |
| *Profitability* | 0.7268 | 0.0685 | *** | 0.7315 | 0.0682 | *** |
| *Leverage* | −0.2918 | 0.0380 | *** | −0.2919 | 0.0378 | *** |
| *China* | −35.3624 | 7.2214 | *** | −36.1927 | 7.2665 | *** |
| *EU* | 25.8427 | 7.7074 | *** | 22.3619 | 7.6989 | *** |
| *Japan* | 15.0954 | 7.3250 | ** | 8.1750 | 7.5865 | |
| *Northamerica* | −10.0927 | 7.0901 | | −6.4053 | 7.0823 | |
| Firm fixed effects | | Yes | | | Yes | |
| Year fixed effects | | Yes | | | Yes | |
| *N* | | 4231 | | | 4231 | |
| Adjusted R$^2$ | | 0.8126 | | | 0.8152 | |

**Table 6.** Industry analysis. The dependent variable is economic sustainability (*Ecnscore*). The models are fixed effects panel regression models (robust standard errors). Model 5 focuses on the aggregate ESG score, while model 6 focuses on the single dimensions of ESG. The industry variable is a dummy equal to 1 if the firm is technological and zero otherwise. For every independent variable, we report the coefficient, the standard error, and the statistical significance: *** for *p*-value < 0.01, ** for *p* < 0.05 and * for *p* < 0.1. Data are collected from 2013 to 2017. Data source: EC JRCand Refinitiv.

| | Model 5 | | | Model 6 | | |
|---|---|---|---|---|---|---|
| | coeff. | se | sig | coeff. | Se | sig |
| *constant* | −32.4684 | 19.3043 | * | −24.0527 | 19.2630 | |
| *MPI* | 0.0492 | 0.0248 | ** | 0.0478 | 0.0247 | * |
| *Esgscore* | 0.5085 | 0.0322 | *** | | | |
| *Envscore* | | | | 0.0569 | 0.0286 | ** |
| *Socscore* | | | | 0.3314 | 0.0266 | *** |
| *Cgvscore* | | | | 0.0752 | 0.0287 | *** |
| *Size* | 3.8767 | 1.0890 | *** | 3.6379 | 1.0826 | *** |
| *Profitability* | 0.7288 | 0.0685 | *** | 0.7337 | 0.0682 | *** |
| *Leverage* | −0.2926 | 0.0380 | *** | −0.2924 | 0.0378 | *** |
| *Tech* | 20.9123 | 7.5049 | *** | 14.6827 | 7.5226 | * |
| Firm fixed effects | | Yes | | | Yes | |
| Year fixed effects | | Yes | | | Yes | |
| *N* | | 4246 | | | 4246 | |
| Adjusted R$^2$ | | 0.8124 | | | 0.8150 | |

## 5. Conclusions

The purpose of this work is an empirical investigation of the relationship between innovation and economic sustainability at the corporate level. On the one hand, the reasons under this work are the strong public consideration of sustainability since the beginning of

the new millennium and, on the other hand, the wide-ranging consensus on the fact that innovation turns out to be positive for growth. Based on our findings, innovation should be considered the interaction or combination of the financial commitments or efforts (R&D investments) supporting a firm's growth opportunities. From this point of view, a company is perceived as successfully innovative if it can combine R&D expenses with the real growth opportunities recognized by the market, according to a principle of cost-effectiveness of investments or specific managerial abilities supporting economic sustainability.

The main finding concerns a positive association between economic sustainability and our market-perceived innovation (MPI) proxy, confirming a virtuous paradigm as achievable. Therefore, at the microeconomic level, to improve their economic sustainability, companies are encouraged to increase their R&D investments in their growth opportunities. At the macroeconomic level, the policies should be mostly driven by incentives for R&D expenditures in line with the real opportunities and with measurable economic sustainability targets. The results also highlight a relevant issue concerning the correlations between the traditional underlying ESG pillars. Economic sustainability is positively affected by environmental, social, and governance issues, but with different impacts. The social pillar exhibits more influence, suggesting that firms should pay attention to these specific issues (however, the environmental and governance pillars should not be negligible). These last results can be very useful for policymakers to identify reform priorities depending on the desirable targets. Last but not least, the intensity of the relationship between innovation and economic sustainability remains stable across countries and industries, although the impact on economic sustainability is also affected by these variables, ceteris paribus. Therefore, investors and policy-makers should compare the relative sustainability performance of firms within a specific industry and country or geographical area.

Given these results, the main policy implications of our study can be summarized as follows: First, policy-makers in charge of sustainable corporate development should become aware of mutual relationships among sustainability pillars, since improving certain pillars may positively or negatively affect others. Second, implementing a well-constructed priority of sustainability pillars is one of the most important steps in building an effective corporate sustainability policy. In doing so, policy-makers require the integration of appropriate methodologies to sort out sustainability priorities by the top-down approach, which involves hierarchical decisions. In other words, policy-makers select the most critical pillars to allocate limited resources for reform priorities. Finally, the reform priorities for corporate sustainability are industry-specific. Policy-makers should focus on homogeneous industries and avoid the generalization of reform priorities across sets of heterogeneous industries.

We are aware that this study may have been influenced by the characteristics of the data and by the methodological choices of analysis. First, recalling that the sample includes worldwide top R&D-spending listed firms, it is reasonable to expect that MPI could reveal a greater impact in discriminating economic sustainability if we extend the analysis to a more heterogeneous sample of companies. We adopt a "cherry-picking" approach to test the role played by MPI, our main hypothesis, considering the most reliable and controlled innovation data and information. Secondly, the time-span covered is limited and does not allow a long-term perspective analysis or a dynamic one. However, results turn out to be stable and might be of interest for their policyimplications.

**Author Contributions:** Conceptualization, L.D.S., B.P. and M.P.; methodology, L.D.S., B.P. and M.P.; software, B.P.; validation, L.D.S., B.P. and M.P.; formal analysis, B.P.; investigation, L.D.S., B.P. and M.P.; resources, L.D.S.; data curation, L.D.S.; writing—original draft preparation, L.D.S., B.P. and M.P.; writing—review and editing, L.D.S., B.P. and M.P.; visualization, L.D.S., B.P. and M.P.; supervision, L.D.S., B.P. and M.P.; project administration, L.D.S.; funding acquisition, B.P. and M.P. All authors have read and agreed to the published version of the manuscript.

**Funding:** This research received no external funding.

**Institutional Review Board Statement:** Not applicable.

**Informed Consent Statement:** Informed consent was obtained from all subjects involved in the study.

**Data Availability Statement:** We use private sources of data and consistent with the contracts entered into by our academic institutions we are not licensed to distribute such information. We refer readers to consult the sources cited and their public availability.

**Acknowledgments:** We acknowledge the use of EU R&D Scoreboard data publicly available at the EC JRC webpage (https://ec.europa.eu/jrc/en) (accessed on 15 October 2019).

**Conflicts of Interest:** The authors declare no conflict of interest.

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
