# Peer review of "Economic Sustainability, Innovation, and the ESG Factors: An Empirical Investigation"

_sustainability, doi:10.3390/su14042270_

Round 1

Reviewer 1 Report

This is a well structured and thought out paper with a generally strong methodology and justification for hypotheses. Overall therefore the paper is well written and is recommended for publication. Some suggestions for minor changes would include:

  1. The title does not reflect the attention given to the social and environmental aspects of sustainability. This can be revised perhaps?
  2. the literature on sustainability and innovation is extensive without sufficient exploration of the environmental, social, and governance aspects on page 5. The earlier part therefore on page 4 can be shortened or made more concise to allow for greater depth and exploration of the literature on the same. 
  3. sample is given but without details on the data collection process and description of the sample. This can be included to strengthen that discussion.

Reviewer 2 Report

The layout of the manuscript is correct.
Hypothesis objective and choice of statistical methods correct.
Statistical analysis comprehensive.
The literature review should be enriched with the latest entries (2020,2021, 2022 - if there are any)

Do the Authors identify the negative effects of innovation on sustainable development? If yes, please put them in last part of article. Your own observations are an integral part of your research and add great value to the manuscript.
Check editorial and technical requirements of your manuscript

Reviewer 3 Report

The paper entitled “Economic Sustainability and Innovation: An Empirical Investigation” focuses on an interesting research topic based on the relationship between economic sustainability and innovation. However, the authors must proceed to some improvements, corrections, and some clarifications.

My comments on improvements and corrections to be made and clarifications are:

  1. In the Keywords add the word Innovation.
  2. In JEL codes, and based on what is written in lines 37-41, the Jel codes of sustainable development or sustainability must be included.
  3. in lines 31 and 32 are presents two questions to be answered, but on lines 30-31 is mentioned: “to answer a fundamental question”.
  4. The sentences in the second paragraph of the Introduction must be properly justified and sustained in the literature review. In line 44, is mentioned a “pioneering perspective”, this must be sustained by the literature review that supports it.

     5. In lines 48-53 are cited direct citations, so please insert the respective            pages on the source of Brown et al. (1987).

  1. Several times in the paper are mentioned “recently” or “more recently”, please avoid this. Cite the source date. In many of the cases, the authors say “recently” and cited in the paragraph sources of 2014, 2016, or previous to this years.
  2. The authors must review the text in order to be more objective, highlighting more clarity to the text and sustaining properly the sentences in the literature review. The literature review also must be updated, especially considering that one of the justifications of the motivations of the study is the “increasing worldwide attention to “sustainability” based on the Sustainable Development Goals adopted by United Nation in 2015. The Databases used also must be referred to, as well as the link from where the data was taken.
  3. The paragraph, in lines 123-131, should be removed to the conclusions.
  4. in line 138, the acronym CSR must be presented for the first time by extent. As well as the first time R&D is mentioned in line 83.
  5. In line 164, is cited “in the literature “, please cite the authors to whom you are referring. As well as in line 173 when “several empirical studies” is written.
  6. in the studies mentioned in lines 176-184, please insert some studies of the last years 2019-2022.
  7. in line 196, what do the authors mean by “true innovation”?
  8. Review carefully the Data and Methodology. The model must be reviewed: a comma is missing, the notation is not correct (is used d1, d2… but is described di). The variables in the model are more clearly described and presented. The choice of the fixed panel date is not properly justified, a Hausman test should be applied to the data to confirm this option, especially considering that the dynamic changes during time.

          In table 1, add the description of which are the independent and the dependent variables. And add a third column relative to the data source. In table 2 add at the end the legend of the statistical significance level of the *s. In panel C of Table 3 is better for the reader to order the countries by alphabetic order.

  1. The empirical results must be re-written and based properly in the literature. In lines, 336-338 is said that the firms are oriented to the innovation of products, processes, and services, why? Innovation was not properly defined.

         In line 345 in mentioned “literature”, please cite the authors and                     sources.

         Some words like “appears” to refer to the results are not objective.                 Please use objective words.

  1. In the conclusions, review carefully the 1st paragraph. How is tested the positive effect of sustainability on productivity and employment? It is not tested, so it is not possible to conclude this with the obtained findings. The last sentence of this paragraph should be sustained by a proper literature source.  In line 412, are mentioned” with different impacts”, explain these different impacts according to the obtained findings.

Reviewer 4 Report

The article is interesting and generally, it deserves to be published with some revisions that are suggested below:

  1. Would you please explain what is the interaction turns out to influence economic sustainability positively which you said in the Abstract?
  2. Can you state what the different pillars of sustainability are in the Abstract? and improve what certain pillars may cause a positive or negative effect on the other pillars?
  3. You may try to state three results after your study in the Abstract, e.g. the second: implementing a well-constructed priority of sustainability pillars is one of the most important steps in building an effective corporate sustainability policy in short in Abstract.
  4. You need to describe what the results are since you have the Hypothesis of H1, H2a to H3c.
  5. Would you please refer to more recent literature e.g. from 2019-2021 since your references are a bit old?

Reviewer 5 Report

The article addresses an important issue of the relationship between sustainable development and innovation. It is adapted to the profile of the journal and takes into account international research. These are the key values of your publication. Nevertheless, it also contains a lot of understatements and ambiguities that need to be supplemented and explained. 1. Literature is insufficient. The authors talk about sustainability, but write very little about innovation and how innovation is assessed and measured. 2. The selection of the test sample is not explained. The authors also do not give any limitations resulting from the selection made. 3. The methodology does not clearly indicate how innovation was measured, but how the social and environmental dimensions of sustainability. Above the issues need to be completed and explained. 

Round 2

Reviewer 4 Report

Thank you for your attention to revise the article that will be easy to understand for reader.

Reviewer 5 Report

The authors made a great effort to improve the article. Thank You. The article could be published in present form.